# Distance-Based Network Recovery under Feature Correlation

**David Adametz, Volker Roth**
Department of Mathematics and Computer Science
University of Basel, Switzerland
{david.adametz,volker.roth}@unibas.ch

## Abstract

We present an inference method for Gaussian graphical models when only pairwise distances of $n$ objects are observed. Formally, this is a problem of estimating an $n \times n$ covariance matrix from the Mahalanobis distances $d_{\mathrm{MH}}(\boldsymbol{x}_i, \boldsymbol{x}_j)$, where object $\boldsymbol{x}_i$ lives in a latent feature space. We solve the problem in fully Bayesian fashion by integrating over the Matrix-Normal likelihood and a Matrix-Gamma prior; the resulting Matrix-T posterior enables network recovery even under strongly correlated features. Hereby, we generalize *TiWnet* [19], which assumes Euclidean distances with strict feature independence. In spite of the greatly increased flexibility, our model *neither* loses statistical power *nor* entails more computational cost. We argue that the extension is highly relevant as it yields significantly better results in both synthetic and real-world experiments, which is successfully demonstrated for a network of biological pathways in cancer patients.

## 1 Introduction

In this paper we introduce the *Translation-invariant Matrix-T* process (*TiMT*) for estimating Gaussian graphical models (GGMs) from pairwise distances. The setup is particularly interesting, as many applications only allow distances to be observed in the first place. Hence, our approach is capable of inferring a network of *probability distributions*, of *strings*, *graphs* or *chemical structures*. We begin by stating the setup of classical GGMs: The basic building block is matrix $\widetilde{X} \in \mathbb{R}^{n \times d}$ which follows the Matrix-Normal distribution [8]

$$\widetilde{X} \sim \mathcal{N}(M, \Psi \otimes I_d). \tag{1}$$

The goal is to identify $\Psi^{-1}$, which encodes the desired dependence structure. More specifically, two objects (= rows) are conditionally independent given all others if and only if $\Psi^{-1}$ has a corresponding zero element. This is often depicted as an undirected graph (see Figure 1), where the objects are vertices and (missing) edges represent their conditional (in)dependencies.

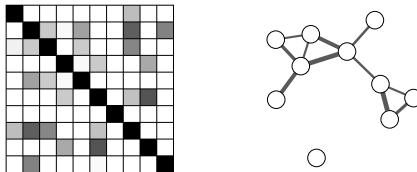

Figure 1: Precision matrix $\Psi^{-1}$ and its interpretation as a graph (self-loops are typically omitted).

Prabhakaran et al. [19] formulated the *Translation-invariant Wishart Network* (*TiWnet*), which treats $\widetilde{X}$ as a latent matrix and only requires their squared Euclidean distances $D_{ij} = d_{\mathrm{E}}(\widetilde{\boldsymbol{x}}_i, \widetilde{\boldsymbol{x}}_j)^2$, where

$\widetilde{\boldsymbol{x}}_i \in \mathbb{R}^d$ is the $i$th row of $\widetilde{X}$. Also, $S_{\mathrm{E}} = \widetilde{X}\widetilde{X}^\top$ refers to the $n \times n$ inner-product matrix, which is linked via $D_{ij} = S_{\mathrm{E},ii} + S_{\mathrm{E},jj} - 2\,S_{\mathrm{E},ij}$. Importantly, the transition to distances implies that means of the form $M = \mathbf{1}_n \boldsymbol{w}^\top$ with $\boldsymbol{w} \in \mathbb{R}^d$ are not identifiable anymore. In contrast to the above, we start off by assuming a matrix

$$X := \widetilde{X}\Sigma^{\frac{1}{2}} \sim \mathcal{N}(M,\,\Psi \otimes \Sigma), \qquad (2)$$

where the columns (= features) are correlated as defined by $\Sigma \in \mathbb{R}^{d \times d}$. Due to this change, the inner-product becomes $S_{\mathrm{MH}} = XX^\top = \widetilde{X}\Sigma\widetilde{X}^\top$. If we directly observed $X$ as in classical GGMs, then $\Sigma$ could be removed to recover $\widetilde{X}$, however, in the case of distances, the impact of $\Psi$ and $\Sigma$ is inevitably mixed. A suitable assumption is therefore the squared *Mahalanobis* distance

$$D_{ij} = d_{\mathrm{MH}}(\boldsymbol{x}_i, \boldsymbol{x}_j)^2 = (\widetilde{\boldsymbol{x}}_i - \widetilde{\boldsymbol{x}}_j)^\top \Sigma (\widetilde{\boldsymbol{x}}_i - \widetilde{\boldsymbol{x}}_j), \qquad (3)$$

which dramatically increases the degree of freedom for inference about $\Psi$. Recall that in our setting only $D$ is observed and the following is latent: $d$, $X$, $\widetilde{X}$, $S := S_{\mathrm{MH}}$, $\Sigma$ and $M = \mathbf{1}_n\boldsymbol{w}^\top$.

The main difficulty comes from the inherent mixture effect of $\Psi$ and $\Sigma$ in the distances, which blurs or obscures what is relevant in GGMs. For example, if we naively enforce $\Sigma = I_d$, then all of the information is solely attributed to $\Psi$. However, in applications where the true $\Sigma \neq I_d$, we would consequently infer false structure, up to a degree where the result is completely mislead by feature correlation.

In pure Bayesian fashion, we specify a prior belief for $\Sigma$ and average over all realizations weighted by the Gaussian likelihood. For a conjugate prior, this leads to the Matrix-T distribution, which forms the core part of our approach. The resulting model generalizes *TiWnet* and is flexible enough to account for arbitrary feature correlation.

In the following, we briefly describe a practical application with all the above properties.

**Example: A Network of Biological Pathways** Using DNA microarrays, it is possible to measure the expression levels of thousands of genes in a patient simultaneously, however, each gene is highly prone to noise and only weakly informative when analyzed on its own. To solve this problem, the focus is shifted towards *pathways* [5], which can be seen as (non-disjoint) groups of genes that contribute to high-level biological processes. The underlying idea is that genes exhibit visible patterns only when paired with *functionally related* entities. Hence, every pathway has a characteristic *distribution* of gene expression values, which we compare via the so-called *Bhattacharyya* distance [2, 11]. Our goal is then to derive a network between pathways, but what if the patients (= features) from whom we obtained the cells were correlated (sex, age, treatment, ...)?

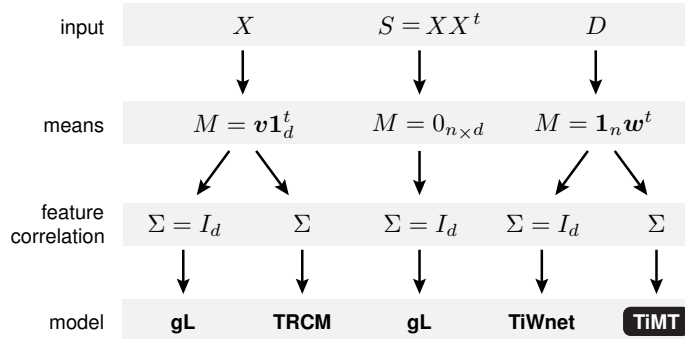

Figure 2: The big picture. Different assumptions about $M$ and $\Sigma$ lead to different models.

**Related work** Inference in GGMs is generally aimed at $\Psi^{-1}$ and therefore every approach relies on Eq. (1) or (2), however, they differ in their assumptions about $M$ and $\Sigma$. Figure 2 puts our setting into a larger context and describes all possible configurations in a single scheme. Throughout the paper, we assume there are $n$ objects and an unknown number of $d$ latent features. Since our inputs are pairwise distances $D$, the mean is of the form $M = \mathbf{1}_n\boldsymbol{w}^\top$, but at the same time, we do not

impose any restriction on $\Sigma$. A complementary assumption is made in *TiWnet* [19], which enforces strict feature independence.

For the models based on matrix $X$, the mean matrix is defined as $M = \boldsymbol{v}\mathbf{1}_d^\top$ with $\boldsymbol{v} \in \mathbb{R}^n$. This choice is neither better nor worse—it does not rely on pairwise distances and hence addresses a different question. By further assuming $\Sigma = I_d$, we arrive at the *graphical LASSO* (*gL*) [7] that optimizes the likelihood under an $L1$ penalty. The *Transposable Regularized Covariance Model* (*TRCM*) [1] is closely related, but additionally allows arbitrary $\Sigma$ and alternates between estimating $\Psi^{-1}$ *and* $\Sigma^{-1}$. The basic configuration for $S$, $M = 0_{n \times d}$ and $\Sigma = I_d$, also leads to the model of *gL*, however this rarely occurs in practice.

## 2   Model

On the most fundamental level, our task deals with incorporating invariances into the Gaussian model, meaning it must not depend on any unrecoverable feature information, i.e. $\Sigma$, $M = \mathbf{1}_n \boldsymbol{w}^\top$ (vanishes for distances) and $d$. The starting point is the log-likelihood of Eq. (2)

$$\ell(W, \Sigma, M\,;X) = \tfrac{d}{2}\log|W| - \tfrac{n}{2}\log|\Sigma| - \tfrac{1}{2}\mathrm{tr}\big(W(X-M)\Sigma^{-1}(X-M)^\top\big), \qquad (4)$$

where we used the shorthand $W := \Psi^{-1}$. In the literature, there exist two conceptually different approaches to achieve invariances: the first is the *classical marginal likelihood* [12], closely related to the *profile likelihood* [16], where a nuisance parameter is either removed by a suitable statistic or replaced by its corresponding maximum likelihood estimate [9]. The second approach follows the *Bayesian marginal likelihood* by introducing a prior and integrating over the product. Hereby, the posterior is a weighted average, where the weights are distributed according to prior belief. The following sections will discuss the required transformations of Eq. (4).

### 2.1   Marginalizing the Latent Feature Correlation

#### 2.1.1   Classical Marginal Likelihood

Let us begin with the attempt to remove $\Sigma$ by explicit reconstruction, as done in McCullagh [13]. Computing the derivative of Eq. (4) with respect to $\Sigma$ and setting it to zero, we arrive at the maximum likelihood estimate $\widehat{\Sigma} = \tfrac{1}{n}(X-M)^\top W(X-M)$, which leads to

$$\ell(W, M\,;X, \widehat{\Sigma}) = \tfrac{d}{2}\log|W| - \tfrac{n}{2}\log|\widehat{\Sigma}| - \tfrac{1}{2}\mathrm{tr}(W(X-M)\widehat{\Sigma}^{-1}(X-M)^\top) \qquad (5)$$

$$= \tfrac{d}{2}\log|W| - \tfrac{n}{2}\log|W(X-M)(X-M)^\top|. \qquad (6)$$

Eq. (6) does not depend on $\Sigma$ anymore, however, note that there is a hidden implication in Eq. (5): $\widehat{\Sigma}^{-1}$ only exists if $\widehat{\Sigma}$ has full rank, or equivalently, if $d \leq n$. Further, even $d = n$ must be excluded, since Eq. (6) would become independent of $X$ otherwise. McCullagh [13] analyzed the Fisher information for varying $d$ and concluded that this model is "a complete success" for $d \ll n$, but "a spectacular failure" if $d \to n$. Since distance matrices typically require $d \geq n$, the approach does not qualify.

#### 2.1.2   Bayesian Marginal Likelihood

Iranmanesh et al. [10] analyzed the Matrix-Normal likelihood in Eq. (4) in conjunction with an *Inverse Matrix-Gamma* (*IMG*) prior—the latter being a generalization of an inverse Wishart prior. It is denoted by $\Sigma \sim \mathrm{IMG}(\alpha, \beta, \Omega)$, where $\alpha > \tfrac{1}{2}(d-1)$ and $\beta > 0$ are shape and scale parameters, respectively. $\Omega$ is a $d \times d$ positive-definite matrix reflecting the expectation of $\Sigma$. This combination leads to the so-called *(Generalized) Matrix T-distribution*[1] $X \sim \mathcal{T}(\alpha, \beta, M, W, \Omega)$ with likelihood

$$\ell(W, M\,;\alpha, \beta, X, \Omega) = \tfrac{d}{2}\log|W| - (\alpha + \tfrac{n}{2})\log|I_n + \tfrac{\beta}{2}W(X-M)\Omega^{-1}(X-M)^\top|. \qquad (7)$$

Compared to the classical marginal likelihood, the obvious differences are $I_n$ and scalar $\beta$, which can be seen as regularization. The limit of $\beta \to \infty$ implies that no regularization takes place

and, interestingly, this likelihood resembles Eq. (6). The other extreme $\beta \to 0$ leads to a likelihood that is independent of $X$. Another observation is that the regularization ensures full rank of $I_n + \frac{\beta}{2} W (X - M) \Omega^{-1} (X - M)^\top$, hence any $d \geq 1$ is valid.

At this point, the Bayesian approach reveals a fundamental advantage: For *TiWnet*, the distance matrix enforced independent features, but now, we are in a position to maintain the full model while adjusting the hyperparameters instead. We propose $\Omega \equiv I_d$, meaning the prior of $\Sigma$ will be centered at independent latent features, which is a common and plausible choice before observing any data. The flexibility ultimately comes from $\alpha$ and $\beta$ when defining a flat prior, which means deviations from independent features are explicitly allowed.

## 2.2 Marginalizing the Latent Means

The fact that we observe a distance matrix $D$ implies that information about the (feature) coordinate system is irrevocably lost, namely $M = \mathbf{1}\boldsymbol{w}^\top$, which is why the means must be marginalized. We briefly discuss the necessary steps, but for an in-depth review please refer to [19, 14, 17]. Following the classical marginalization, it suffices to define a projection $L \in \mathbb{R}^{(n-1) \times n}$ with property $L\mathbf{1}_n = \mathbf{0}_{n-1}$. In other words, all biases of the form $\mathbf{1}_n \boldsymbol{w}^\top$ are mapped to the nullspace of $L$. The Matrix T-distribution under affine transformations [10, Theorem 3.2] reads $LX \sim \mathcal{T}(\alpha, \beta, LM, L\Psi L^\top, \Omega)$ and in our case ($\Omega = I_d$, $LM = L\mathbf{1}_n \boldsymbol{w}^\top = 0_{(n-1) \times d}$), we have

$$\ell(\Psi; \alpha, \beta, LX) = -\tfrac{d}{2} \log |L\Psi L^\top| - (\alpha + \tfrac{n-1}{2}) \log |I_n + \tfrac{\beta}{2} L^\top (L\Psi L^\top)^{-1} L X X^\top|. \quad (8)$$

Note that due to the statistic $LX$, the likelihood is constant over all $X$ (or $S$) mapping to the same $D$. As we are not interested in any specifics about $L$ other than its nullspace, we replace the image with the kernel of the projection and define matrix $Q := I_n - (\mathbf{1}_n^\top W \mathbf{1}_n)^{-1} \mathbf{1}_n \mathbf{1}_n^\top W$. Using the identity $QSQ^\top = -\tfrac{1}{2} QDQ^\top$ and $Q^\top W Q = WQ$, we can finally write the likelihood as

$$\ell(W; \alpha, \beta, D, \mathbf{1}_n) = \tfrac{d}{2} \log |W| - \tfrac{d}{2} \log(\mathbf{1}_n^\top W \mathbf{1}_n) - (\alpha + \tfrac{n-1}{2}) \log |I_n - \tfrac{\beta}{4} WQD|, \quad (9)$$

which accounts for arbitrary latent feature correlation $\Sigma$ *and* all mean matrices $M = \mathbf{1}_n \boldsymbol{w}^\top$.

In hindsight, the combination of Bayesian and classical marginal likelihood might appear arbitrary, but both strategies have their individual strengths. Mean matrix $M$, for example, is limited to a single direction in an $n$ dimensional space, therefore the statistic $LX$ represents a convenient solution. In contrast, the rank-$d$ matrix $\Sigma$ affects a much larger spectrum that cannot be handled in the same fashion—ignoring this leads to a degenerate likelihood as previously shown. The problem is only tractable when specifying a prior belief for Bayesian marginalization. On a side note, the Bayesian posterior includes the classical marginal likelihood for the choice of an improper prior [4], which could be seen in the Matrix-T likelihood, Eq. (7), in the limit of $\beta \to \infty$.

## 3 Inference

The previous section developed a likelihood for GGMs that conforms to all aspects of information loss inherent to distance matrices. As our interest lies in the network-defining $W$, the following will discuss Bayesian inference using a Markov chain Monte Carlo (MCMC) sampler.

**Hyperparameters $\alpha$, $\beta$ and $d$** At some point in every Bayesian analysis, all hyperparameters need to be specified in a sensible manner. Currently, the occurrence of $d$ in Eq. (9) is particularly problematic, since (i) the number of latent features is unknown and (ii) it critically affects the balance between determinants. To resolve this issue, recall that $\alpha$ must satisfy $\alpha > \tfrac{1}{2}(d - 1)$, which can alternatively be expressed as $\alpha = \tfrac{1}{2}(vd - n + 1)$ with $v > 1 + \tfrac{n-2}{d}$. Thereby, we arrive at

$$\ell(W; v, \beta, D, \mathbf{1}_n) = \tfrac{d}{2} \log |W| - \tfrac{d}{2} \log(\mathbf{1}_n^\top W \mathbf{1}_n) - \tfrac{vd}{2} \log |I_n - \tfrac{\beta}{4} WQD|, \quad (10)$$

where $d$ now influences the likelihood on a global level and can be used as *temperature* reminiscent of *simulated annealing* techniques for optimization. In more detail, we initialize the MCMC sampler with a small value of $d$ and increase it slowly, until the acceptance ratio is below, say, 1 percent. After that event, all samples of $W$ are averaged to obtain the final network.

Parameter $v$ and $\beta$ still play a crucial role in the process of inference, as they distribute the probability mass across all latent feature correlations and effectively control the scope of plausible $\Sigma$. Upon

**Algorithm 1** One loop of the MCMC sampler
***
**Input:** distance matrix $D$, temperature $d$ and fixed $v > 1 + \frac{n-2}{d}$
**for** $i = 1$ **to** $n$ **do**
    $W^{(p)} \leftarrow W$,     $^{(p)}$ refers to *proposal*
    Uniformly select node $k \neq i$ and sample element $W_{ik}^{(p)}$ from $\{-1, 0, +1\}$
    Set $W_{ki}^{(p)} \leftarrow W_{ik}^{(p)}$ and update $W_{ii}^{(p)}$ and $W_{kk}^{(p)}$ accordingly
    Compute posterior in Eq. (12) and acceptance of $W^{(p)}$
    **if** $u \sim \mathcal{U}(0,1) < $ acceptance **then**
        $W \leftarrow W^{(p)}$
    **end if**
**end for**
Sample proposal $\beta^{(p)} \sim \Gamma(\beta_{\text{shape}}, \beta_{\text{scale}})$
Compute posterior in Eq. (12) and acceptance of $\beta^{(p)}$
**if** $u \sim \mathcal{U}(0,1) < $ acceptance **then**
    $\beta \leftarrow \beta^{(p)}$
**end if**
***

closer inspection, we gain more insight by the variance of the Matrix-T distribution,

$$\frac{2(\Psi \otimes \Omega)}{\beta(v\,d - 2\,n + 1)}, \tag{11}$$

which is maximal when $\beta$ and $v$ are jointly small. We aim for the most flexible solution, thus $v$ is fixed at the smallest possible value and $\beta$ is stochastically integrated out in a Metropolis-Hastings step. A suitable choice is a Gamma prior $\beta \sim \Gamma(\beta_{\text{shape}}, \beta_{\text{scale}})$; its shape and scale must be chosen to be sufficiently flexible on the scale of the distance matrix at hand.

**Priors for $W$**    The prior for $W$ is first and foremost required to be sparse and flexible. There are many valid choices, like *spike and slab* [15] or *partial correlation* [3], but we adapt the two-component scheme of *TiWnet*, which has computational advantages and enables symmetric random walks. The following briefly explains the construction:

Prior $p_1(W)$ defines a symmetric random matrix, where off-diagonal elements $W_{ij}$ are uniform on $\{-1, 0, +1\}$, i.e. an edge with positive/negative weight or no edge. The diagonal is chosen such that $W$ is positive definite: $W_{ii} \leftarrow \epsilon + \sum_{j \neq i} |W_{ij}|$. Although this only allows 3 levels, it proved to be sufficiently flexible in practice. Replacing it with more levels is possible, but conceptually identical. The second component is a Laplacian $p_2(W \mid \lambda) \propto \exp\left(-\lambda \sum_{i=1}^{n} (W_{ii} - \epsilon)\right)$ and induces sparsity. Here, the total number of edges in the network is penalized by parameter $\lambda > 0$. Combining the likelihood of Eq. (10) and the above priors, the final posterior reads:

$$p(W \mid \bullet) = p(D \mid W, \beta, \mathbf{1}_n)\, p_1(W)\, p_2(W \mid \lambda)\, p_3(\beta \mid \beta_{\text{shape}}, \beta_{\text{scale}}). \tag{12}$$

The full scheme of the MCMC sampler is reported in Algorithm 1.

**Complexity Analysis**    The runtime of Algorithm 1 is primarily determined by the repeated evaluation of the posterior in Eq. (12), which would require $\mathcal{O}(n^4)$ in the naive case of fully recomputing the determinants. Every flip of an edge, however, only changes a maximum of 4 elements[2] in $W$, which gives rise to an elegant update scheme building on the QR decomposition.

**Theorem.** *One full loop in Algorithm 1 requires $\mathcal{O}(n^3)$.*

*Proof.* Due to the 3-level prior, there are only 6 possible flip configurations depending on the current edge between object $i$ and $j$ (2 examples depicted here for $i = 1$, $j = 3$):

$$\Delta W := W^{(p)} - W \quad \Leftrightarrow \quad \left\{ \begin{bmatrix} -\mathbf{1} & 0 & +\mathbf{1} \\ 0 & 0 & 0 \\ +\mathbf{1} & 0 & -\mathbf{1} \end{bmatrix}, \dots, \begin{bmatrix} \mathbf{0} & 0 & +\mathbf{2} \\ 0 & 0 & 0 \\ +\mathbf{2} & 0 & \mathbf{0} \end{bmatrix} \right\} \tag{13}$$

An important observation is that $\Delta W$ can solely be expressed in terms of rank-1 matrices, in particular either $\boldsymbol{u}\boldsymbol{v}^\top$ or $\boldsymbol{u}\boldsymbol{v}^\top + \boldsymbol{a}\boldsymbol{b}^\top$. If we know the QR decomposition of $W$, then the decomposition

***
[2]This also holds for more than 3 edge levels.

of $W^{(p)}$ can be found in $\mathcal{O}(n^2)$. Consequently, its determinant is obtained by $\det(QR) = \prod_{i=1}^n R_{ii}$ in $\mathcal{O}(n)$. Our goal is to exploit this property and express both determinants of the posterior as rank-1 updates to their existing QR decompositions. Restating the likelihood, we have

$$\ell(W^{(p)}\,;\,\bullet) = \tfrac{d}{2}\log\underbrace{|W^{(p)}|}_{=:\,\det_1} - \tfrac{d}{2}\log(\mathbf{1}_n^\top W^{(p)}\mathbf{1}_n) - \tfrac{vd}{2}\log\underbrace{|I_n - \tfrac{\beta}{4}W^{(p)}QD|}_{=:\,\det_2}. \qquad (14)$$

Updating $\det_1$ corresponds to either $W^{(p)} = W + \boldsymbol{u}\boldsymbol{v}^\top$ or $W^{(p)} = W + \boldsymbol{u}\boldsymbol{v}^\top + \boldsymbol{a}\boldsymbol{b}^\top$ as explained in Eq. (13), thus leading to $\mathcal{O}(n^2)$. We reformulate $\det_2$ to follow the same scheme:

$$\begin{aligned}
\det_2 = \Big| & I_n - \tfrac{\beta}{4}W\left(I_n - \tfrac{1}{\mathbf{1}_n^\top W\mathbf{1}_n}\mathbf{1}_n\mathbf{1}_n^\top W\right)D \\
& - \tfrac{\beta}{4}\left[\left(\tfrac{1}{\mathbf{1}_n^\top W\mathbf{1}_n} - \gamma\right)W\mathbf{1}_n - \gamma\left((\boldsymbol{v}^\top\mathbf{1}_n)\boldsymbol{u} + (\boldsymbol{b}^\top\mathbf{1}_n)\boldsymbol{a}\right)\right]\left(DW\mathbf{1}_n\right)^\top \\
& - \tfrac{\beta}{4}\left[\boldsymbol{u} - \gamma\left(\mathbf{1}_n^\top\boldsymbol{u}\right)\left(W\mathbf{1}_n + (\boldsymbol{v}^\top\mathbf{1}_n)\boldsymbol{u} + (\boldsymbol{b}^\top\mathbf{1}_n)\boldsymbol{a}\right)\right]\left(D\boldsymbol{v}\right)^\top \\
& - \tfrac{\beta}{4}\left[\boldsymbol{a} - \gamma\left(\mathbf{1}_n^\top\boldsymbol{a}\right)\left(W\mathbf{1}_n + (\boldsymbol{v}^\top\mathbf{1}_n)\boldsymbol{u} + (\boldsymbol{b}^\top\mathbf{1}_n)\boldsymbol{a}\right)\right]\left(D\boldsymbol{b}\right)^\top\Big|.
\end{aligned} \qquad (15)$$

For notational convenience, we defined the shorthand

$$\gamma := \frac{1}{\mathbf{1}_n^\top W^{(p)}\mathbf{1}_n} = \frac{1}{\mathbf{1}_n^\top(W + \boldsymbol{u}\boldsymbol{v}^\top + \boldsymbol{a}\boldsymbol{b}^\top)\mathbf{1}_n} = \frac{1}{\mathbf{1}_n^\top W\mathbf{1}_n + (\mathbf{1}_n^\top\boldsymbol{u})(\boldsymbol{v}^\top\mathbf{1}_n) + (\mathbf{1}_n^\top\boldsymbol{a})(\boldsymbol{b}^\top\mathbf{1}_n)}.$$

Note that the determinant of the first line in Eq. (15) is already known (i.e. its QR decomposition) and the following 3 lines are only rank-1 updates as indicated by parenthesis. Therefore, $\det_2$ is computed in 3 steps, each consuming $\mathcal{O}(n^2)$. For some of the 6 flip configurations, we even have $\boldsymbol{a} = \boldsymbol{b} = \mathbf{0}_n$, which renders the last line in Eq. (15) obsolete and simplifies the remaining terms.

Since the for loop covers $n$ flips, all updates contribute as $n \cdot \mathcal{O}(n^2)$. There is no shortcut to evaluate proposal $\beta^{(p)}$ given $\beta$, thus its posterior is recomputed from scratch in $\mathcal{O}(n^3)$. Therefore, Algorithm 1 has an overall complexity of $\mathcal{O}(n^3)$, which is the same as *TiWnet*. $\qquad\square$

## 4 Experiments

### 4.1 Synthetic Data

We first look at synthetic data and compare how well the recovered network matches the true one. Hereby, the accuracy is measured by the f-score using the edges (positive/negative/zero).

**Independent Latent Features**  Since *TiMT* is a generalization for arbitrary $\Sigma$, it must also cover $\Sigma \equiv I_d$, thus, we generate a set of 100 Gaussian-distributed matrices $X$ with known $W$ and $\Sigma = I_d$, where $n = 30$ and $d = 300$. Next, we add column translations $\mathbf{1}_n\boldsymbol{w}^\top$ with elements in $\boldsymbol{w} \in \mathbb{R}^d$ being Gamma distributed, however these do not enter $D$ by definition. As *TRCM* does not account for column shifts, it is used in conjunction with the *true, unshifted* matrix $X$ (hence *TRCM.u*).

All methods require a regularization parameter, which obviously determines the outcome. In particular, *TiWnet* and *TiMT* use the same, constant parameter throughout all 100 distance matrices and obtain the final $W$ via annealing. Concerning *TRCM* and *gL*, we evaluate each $X$ on a set of parameters and only report the *highest f-score per data set*. This is in strong favor of the competition.

Boxplots of the achieved f-scores and the false positive rates are depicted in Figure 3, left. As can be seen, *TiMT* and *TiWnet* score as high as *TRCM.u without* knowledge of features or feature translations. We omit *gL* from the comparison due to a model mismatch regarding $M$, meaning it will naturally fall short. Instead, the interested reader is pointed to extensive results in [19].

The gist of this experiment is that all methods work well when the model requirements are met. Also, translating the individual features and obscuring them does not impair *TiWnet* and *TiMT*.

**Correlated Latent Features**  The second experiment is similar to the first one ($n = 30$, $d = 300$ and column shifts), but it additionally introduces feature correlation. Here, $\Sigma$ is generated by sampling a matrix $G \sim \mathcal{N}(0_{d \times 5d}, I_d \otimes I_{5d})$ and adding Gamma distributed vector $\boldsymbol{a} \in \mathbb{R}^{5d}$ to randomly selected rows of $G$. The final feature covariance matrix is given by $\Sigma = \tfrac{1}{5d}GG^\top$.

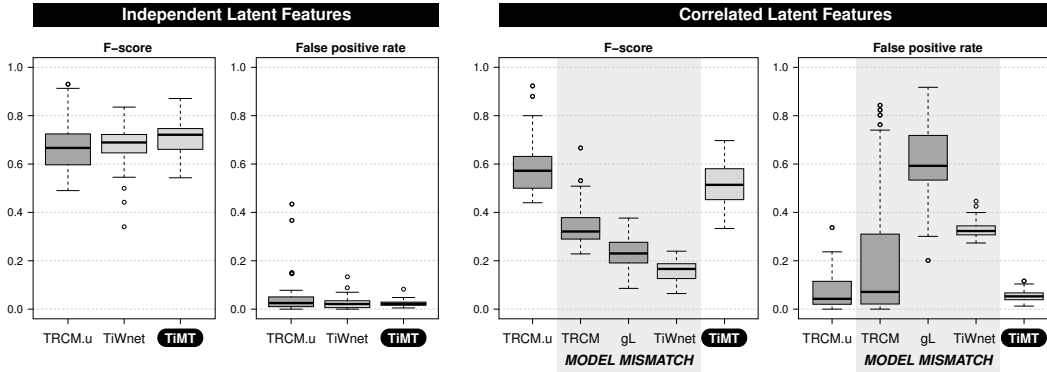

Figure 3: Results for synthetic data. Translations do *not* apply to *TRCM.u*. Models with violated assumptions ($M$ and/or $\Sigma$) are highlighted with gray bars.

Due to the dramatically increased degree of freedom, all methods are impacted by lower f-scores (see Figure 3, right). As expected, *TRCM.u* performs best in terms of f-score, which is based on the unshifted full data matrix $X$ with an individually optimized regularization parameter. *TiMT*, however, follows by a slim margin. On the contrary, *TiWnet* explains the similarities exclusively by adding more (unnecessary) edges, which is reflected in its increased, but strongly consistent false positive rate. This issue leads to a comparatively low f-score that is even below the remaining contenders. Finally, Figure 4 shows an example network and its reconstruction. Keeping in mind the drastic information loss between true $X_{30 \times 300}$ and $D_{30 \times 30}$, *TiMT* performs extremely well.

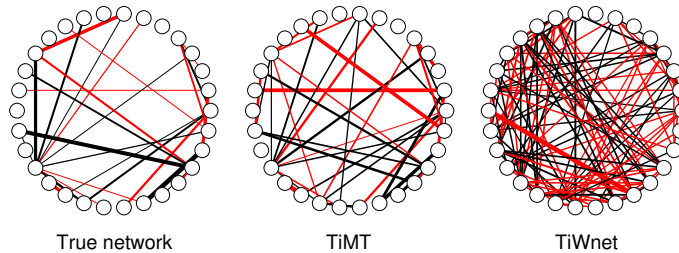

| True network | TiMT | TiWnet |

Figure 4: An example for synthetic data with feature correlation. The network inferred by *TiMT* (center) is relatively close to ground truth (left), however *TiWnet* (right) is apparently mislead by $\Sigma$. Black/red edges refer to $+/-$ edge weight.

## 4.2 Real-World Data: A Network of Biological Pathways

In order to demonstrate the scalability of TiMT, we apply it to the publicly available colon cancer dataset of Sheffer et al. [20], which is comprised of $13\,437$ genes measured across $182$ patients. Using the latest gene sets from the KEGG[3] database, we arrive at $n = 276$ distinct pathways. After learning the mean and variance of each pathway as the distribution of its gene expression values across patients, the Bhattacharyya distances [11] are computed as a $276 \times 276$ matrix $D$. The pathways are allowed to overlap via common genes, thus leading to similarities, however it is unclear how and to what degree the correlation of patients affects the inferred network. For this purpose, we run *TiMT* alongside *TiWnet* with identical parameters for $20\,000$ samples and report the annealed networks in Figure 5. Again, the difference in topology is only due to latent feature correlation. Runtime on a standard $3\,\text{GHz}$ PC was 3:10 hours for *TiMT*, while a naive implementation in $\mathcal{O}(n^4)$ finished after ~20 hours. *TiWnet* performed slightly better at around 3 hours, since the model does not have hyperparameter $\beta$ to control feature correlation.

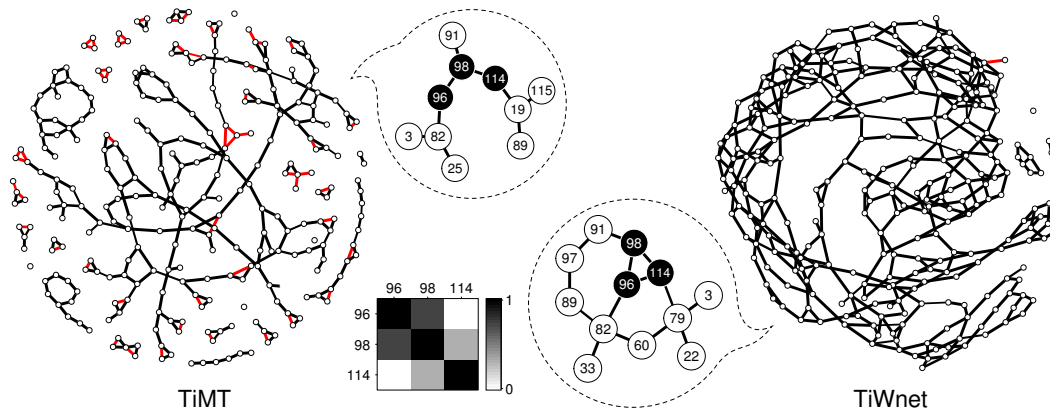

Figure 5: A network of pathways in colon cancer patients, where each vertex represents one pathway. From both results, we extract a subgraph of 3 pathways including all neighbors in reach of 2 edges. The matrix on the bottom shows external information on pathway similarity based on their relative number of protein-protein interactions. Black/red edges refer to $+/-$ edge weight.

Without side information it is not possible to confirm either result, hence we resort to expert knowledge for protein-protein interactions from the BioGRID[4] database and compute the strength of connection between pathways as the number of interactions relative to their theoretical maximum. Using this, we can easily check subnetworks for plausibility (see Figure 5, center): The black vertices 96, 98 and 114 correspond to *base excision repair*, *mismatch repair* and *cell cycle*, which are particularly interesting as they play a key role in DNA mutation. These pathways are known to be strongly dysregulated in colon cancer and indicate an elevated susceptibility [18, 6]. The topology of these 3 pathways for *TiMT* is fully supported by protein interactions, i.e. 98 is the link between 114 and 96 and removing it renders 96 and 98 independent. *TiWnet*, on the contrary, overestimates the network and produces a highly-connected structure contradicting the evidence. This is a clear indicator for latent feature correlation.

## 5    Conclusion

We presented the *Translation-invariant Matrix-T* process (*TiMT*) as an elegant way to make inference in Gaussian graphical models when only pairwise distances are available. Previously, the inherent information loss about underlying features appeared to prevent any conclusive statement about their correlation, however, we argue that neither assumed full independence nor maximum likelihood estimation is reasonable in this context.

Our contribution is threefold: (i) A Bayesian relaxation solves the issue of strict feature independence in GGMs. The assumption is now shifted into the prior, but flat priors are possible. (ii) The approach generalizes *TiWnet*, but maintains the same complexity, thus, there is no reason to retain the simplified model. (iii) *TiMT* for the first time accounts for *all* latent parameters of the Matrix Normal *without* access to the latent data matrix $X$. The distances $D$ are fully sufficient.

In synthetic experiments, we observed a substantial improvement over *TiWnet*, which highly overestimated the networks and falsely attributed all information to the topological structure. At the same time, *TiMT* performed almost on par with *TRCM(.u)*, which operates under hypothetical, optimal conditions. This demonstrates that all aspects of information loss can be handled exceptionally well.

Finally, the network of biological pathways provided promising results for a domain of non-vectorial objects, which effectively precludes all methods except for *TiMT* and *TiWnet*. Comparing these two, the considerable difference in network topology only goes to show that invariance against latent feature correlation is indispensable—especially pertaining to distances.

## Footnotes

[1]Choosing an inverse Wishart prior for $\Sigma$ results in the *standard* Matrix T-distribution, however its variance can only be controlled by an integer. This is why the *Generalized* Matrix T-distribution is preferred.

[3]http://www.genome.jp/kegg/, accessed in May 2014

[4]http://thebiogrid.org, version 3.2

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
