[Reviews · NeurIPS 2014]

Submitted by Assigned_Reviewer_9

Summary: This work proposes a new method for learning the structure of
a Gaussian graphical model. It seeks to account for hidden feature
correlations when determining the structure by modeling the feature
covariance matrix using a Matrix-Gamma prior. In contrast, prior work
such as TiWnet [19] assumes feature independence, which can lead to an
overly-dense graph. Synthetic experiments, as well as experiments in
which networks of genetic pathways are learned, clearly demonstrate
the superiority of the proposed method over TiWnet [19].

Clarity, quality, originality, significance: The exposition is fairly
clear. There seem to be many similarities to the TiWnet work, so,
while the background is a little sparse, most readers' questions can
be answered by referring to [19]. Figure 2 also provides a very nice,
clean comparison to related work, and section 2.1.1's explanation of
why Sigma can't simply be removed by setting the derivative equal to
zero is helpful. While it's not a large theoretical delta over the
TiWnet work, the experiments indicate the practical edge the use of
the Matrix-Gamma prior gives, which testifies to the work's
significance.

More detailed comments:

It seems that there are many similarities between this work and the
TiWnet work that aren't exactly explicit in the paper. For instance,
Algorithm 1 is nearly identical to the algorithm for TiWnet, except
for the beta sampling step. It would be helpful to the reader if
these sort of similarities and differences were more clearly marked.

Please explain in more detail exactly what is done experimentally to
"obtain the final W via annealing". How is the temperature d adjusted
to converge on a final W?

In the experiments, how is lambda for the prior p_2 chosen? Since
this prior is meant to induce sparsity, why not just turn up lambda
for TiWnet? This seems like it might at least partially solve the
main problem observed in the experiments, where TiWnet learns far too
dense of a graph.

It would be interesting to see some discussion of future work and the
deficiencies of the proposed method. Does TiMT have any obvious
weaknesses like TiWnet has? What sort of connections does it fail to
predict and why? Is the fact that the W matrix is only allowed three
levels (+1, -1, 0) something that could be easily improved upon? It
seems like most of the analysis of Algorithm 1's efficiency depends
upon the fact that the update to W can be expressed in terms of a
small number of rank-1 matrices. Perhaps this could be further
leveraged to increase the number of levels in W without significantly
increasing the computational cost.

Thanks very much for including code with this submission. One small
comment --- while the demo_script is easy to parse, it would be
helpful if the timt script had some comments in the code.

Minor notes:

0) "Gaussian Graphical Models" -> "Gaussian graphical models"

1) "our approach is capable to infer" -> "our approach is capable of
inferring"

2) "can be removed to receive" -> "can be removed to recover"

3) "it is possibly to measure" -> "it is possible to measure"

4) "levels of hundred thousands of genes" -> "levels of hundreds of
thousands of genes"

5) "’a complete success’" -> "'a complete success'"

6) "’a spectacular failure’" -> "'a spectacular failure'"

7) Copy "Black/red edges refers to +/− edge weight." from the caption
of Figure 5 up to the caption of Figure 4.

8) "Bhattacharyya distance[11] are computed as" -> "Bhattacharyya
distances [11] are computed as"

9) "and to which degree" -> "and to what degree"

10) "cannot be verified, however, there" -> "cannot be verified;
however, there"

11) "information loss about underlying features" -> "information lost
about underlying features"

12) Supplement: "we are interested to find" -> "we are interested in
finding"
Summary: This work proposes a new method for learning the structure of a
Gaussian graphical model that, in contrast to previous work, attempts
to account for feature correlations. While this isn't a large
theoretical delta over [19], the new method has a clear empirical
advantage.

Submitted by Assigned_Reviewer_28

This paper considers inference of Gaussian Graphical Model when only pairwise distances of observations are observed. The paper proposes a marginal likelihood (TIMT) obtained by marginalizing the latent feature correlation and the latent mean. This likelihood is embedded within a Bayesian framework and they propose a MCMC algorithm obtain posterior estimates of the W matrix. Some of the technical challenges in the inference arise from : the number of latent features is unknown when only the distances are observed but the posterior depends explicility on d, and each loop of the MCMC would require O(n^4) time if computed naively. The first is solved by a reparameterization that leads to the d scaling the likelihood and the second is solved by using low-rank updates for the MCMC proposal.
Simulations show that this framework performs favorably compared to approaches such as TiWNet.

The paper is a useful contribution to methods for GGM inference -- the ability to work with distances is useful in a variety of applications. The paper is clearly written.

Comments
1. There are aspects of the inference that are not clear to me. How do they set d and v in practice? What W were used to simulate data ? How was the final W obtained ? How many iterations of MCMC were needed ?
2. How does TIMT compare to other methods in terms of computational efficiency ?
3. It appears that the comparison only uses the zero,non-zero values of inferred W. How well do the methods do in inferring the quantitative values of W ?
4. It would be more informative to show Precision-Recall curves for the methods rather than report a F-score.
5. It would also be useful to repeat these comparisons for varying n/d ratios.

Summary: The proposed approach generalizes several proposed inference algorithms for GGMs and is shown to perform favorably in a simulation.

Submitted by Assigned_Reviewer_43

This paper introduces a new method for estimating graphical model structure from pairwise distances. The method differs from existing methods by explicitly modeling feature correlation --- that is, it decomposes the covariance between data points into an overall dependence structure term and a separate feature correlation term. This differs from previous work, which assumed independence between different dimensions of the data. A prior is placed on the feature correlation matrix, which allows the feature correlation matrix itself to be marginalized out analytically; inference over the latent structure (a precision matrix) is then performed via MCMC. Updates to the precision matrix during MCMC are treated numerically as rank-one updates to a QR decomposition, keeping the overall runtime of a single sweep through the data O(n^3) despite the need to compute the determinant of an n-by-n matrix for each of the n data points. The experiments demonstrate a remarkable improvement over existing data, both in synthetic and real-world examples.

This paper is exceptionally well written and well-motivated. The Bayesian extension to existing methods is placed clearly into the context of related work. The methods are clearly described and replicable; additionally, full source code is provided in the supplemental material.

The estimated structures in the example applications represent a clear order-of-magnitude improvement over existing approaches which do not take feature correlation into account. These experimental results suggest that this should be the new standard method by which one estimates dependence structures in Gaussian graphical models in cases where the dimensionality of the data is large relative to the number of data points.
Summary: This well-written paper introduces a new method for estimating dependence structure from pairwise distances, which represents a dramatic improvement over existing methods while retaining the same overall computational complexity.
Author Feedback
Author rebuttal: Thank you to all reviewers for their valuable, in-depth feedback. We try to answer all questions as concisely as possible.

1. How to set d and v in practice? How is W inferred? (Rev9 & 28)
Both d and v affect the variance of the distribution and hereby we control how the sampler explores the space of all W. In particular, we fix v and only vary d.

As for v (originally the width of the Sigma prior), small values allow the model to be flexible, therefore we choose the smallest possible option (lower bound depending on d, e.g. n = d = 100, v !> 1.98). For large v, the prior strongly peaks at Omega = I_d and TiMT behaves exactly like TiWnet.

d is the unknown number of features, but we interpret it as an annealing parameter, since it occurs as a factor for all terms in the log-likelihood (= an exponent in the likelihood). Typically, the sampler is initialized with a small d (e.g. d = n) such that the variance is maximal. At this stage, flips are very likely to be accepted, even when they lead to a worse configuration. By slowly increasing d, less and less proposals are accepted until it eventually comes to a halt. At this point, we obtain the final W.

* What W were used in simulations? (Rev28)
The true W has a sparse structure and continuous weights. Its structure is generated from a binomial with 1 trial and pareto-distributed probabilities. 50% of 1's (uniformly selected) are set to -1. Finally, the weights are given by a gamma distribution with scale and shape = 2. The whole generating scheme can be found in the demo script, file "timt.R", function "sample.gauss.wishart".

* How many iterations of MCMC were used in the experiments? (Rev28)
The synthetic experiments ran 5.000 iterations for each of the 100 networks, the pathway network consumed 20.000 as described.

2. Efficiency of TiMT compared to other methods. (Rev28)
Our problem setting is motivated by the limited information inherent to distance matrix D, thus neither gLASSO nor TRCM are actual competitors. This only leaves TiWnet, which has the same requirements as TiMT, i.e. O(n^3).

3. Prior for W: quantitative performance? Improvement by k levels? (Rev9 & 28)
A 3-level prior is the simplest discretization to allow inference about topology and edge signs. If detailed edge weights are needed, 3 levels can easily be increased to k levels _without_ affecting the rank-1 updates. Note that the scale of a distance matrix is typically without information, hence, the prior must live on a fixed range of values. Also, oscillating behavior with beta must be avoided--beta controls the level of regularization and is therefore linked to the scale.

On the downside, the sampler will become slower with more levels, as it must explore a larger space of possible W. Thus, there is a tradeoff between speed of the sampler and better approximation of W. One could try to decouple the proposals, i.e. first sample edge/no edge, then sample weight (if needed). Special case: If the structure is simple, then increasing levels has only little impact on speed.

4. Precision-recall instead of F-score. (Rev28)
A boxplot over F-scores was chosen to give the best picture for performance across a variety of 100 data sets. A PR curve would be meaningful for parameter tuning on a single data set, e.g. the sparsity level (which we tuned beforehand and fixed during all trials).

It is possible to replace each F-score boxplot by 2 boxplots (P & R), but this would lead to crowded figures. Also, note that gLASSO and TRCM had the advantage of an F-score-optimized sparsity parameter per data set. Optimizing it by 2 criteria would be cumbersome.

5. Experiments with different n/d ratios. (Rev28)
For generating data: (n = 30, d = 300) was chosen to receive a full rank matrix D with visible and sufficiently random feature correlation. We tested many combinations and found out that all methods suffer heavily for small d. This is to be expected, because when the variance is large, a sample S (or D) can differ completely from its underlying covariance matrix. Also, the results become almost random (across all methods!). Conversely, when d is too large, the variance is zero and the task becomes too easy. Thus, we chose a middle ground where the problem is challenging (F-score approx. 0.5) and the differences between methods are visible.

For inference: d is interpreted as real-valued annealing temperature. When slowly increasing d, the acceptance of flips will eventually stop (see question 1), but this particular value does not carry any meaning for the interpretation of the network.

* How is sparsity parameter lambda chosen? TiWnet graph too dense--a matter of lambda tuning? (Rev9)
All 4 methods require a user-defined sparsity level, thus an "appropriate" choice depends on background knowledge or an expert. In the synthetic experiments, it was chosen such that the F-score is maximized.

Forcing TiWnet to produce a more sparse topology might appear logical at first, but this will equally remove false _and_ true positive edges. In TiWnet's model, feature independence is hard-coded by design, hence it is forced to explain all observations by structure only. It cannot distinguish these two sources of input.

* Limitations of TiMT? (Rev9)
For the first time, all parameters of the Matrix-Normal were considered, thus the foundation is most flexible. Still, a distance matrix obscures many vital aspects of the data, which--technically speaking--makes a small portion of structure irrecoverable. While TiMT requires even less information than TiWnet, the question is where the line can be drawn such that inference is still feasible.

As for future work, handling missing values would be of interest to us.

* Typos and corrections. (Rev9)
All suggestions are greatly welcome and have already been integrated.